# Effects of Dibutylphthalate and Steroid Hormone Mixture on Human Prostate Cells

**DOI:** 10.3390/ijms241814341

**Published:** 2023-09-20

**Authors:** Aldo Mileo, Teresa Chianese, Gianluca Fasciolo, Paola Venditti, Anna Capaldo, Luigi Rosati, Maria De Falco

**Affiliations:** 1Department of Biology, University Federico II of Naples, Via Cinthia 26, 80126 Naples, Italy; aldo.mileo@unina.it (A.M.); teresa.chianese2@unina.it (T.C.); gianluca.fasciolo@unina.it (G.F.); paola.venditti@unina.it (P.V.); anna.capaldo@unina.it (A.C.); luigi.rosati@unina.it (L.R.); 2CIRAM, Centro Interdipartimentale di Ricerca “Ambiente”, University Federico II of Naples, Via Mezzocannone 16, 80134 Naples, Italy; 3National Institute of Biostructures and Biosystems (INBB), Viale delle Medaglie d’Oro 305, 00136 Rome, Italy

**Keywords:** phthalates, di-n-butyl phthalate (DBP), endocrine-disrupting chemicals (EDCs), prostate gland, steroid receptors, estrogens, androgens

## Abstract

Phthalates are a family of aromatic chemical compounds mainly used as plasticizers. Among phthalates, di-n-butyl phthalate (DBP) is a low-molecular-weight phthalate used as a component of many cosmetic products, such as nail polish, and other perfumed personal care products. DBP has toxic effects on reproductive health, inducing testicular damage and developmental malformations. Inside the male reproductive system, the prostate gland reacts to both male and female sex steroids. For this reason, it represents an important target of endocrine-disrupting chemicals (EDCs), compounds that are able to affect the estrogen and androgen signaling pathways, thus interfering with prostate homeostasis and inducing several prostate pathologies. The aim of this project was to investigate the effects of DBP, alone and in combination with testosterone (T), 17β-estradiol (E2), and both, on the normal PNT1A human prostate cell-derived cell line, to mimic environmental contamination. We showed that DBP and all of the tested mixtures increase cell viability through activation of both estrogen receptor α (ERα) and androgen receptor (AR). DBP modulated steroid receptor levels in a nonmonotonic way, and differently to endogenous hormones. In addition, DBP translocated ERα to the nucleus over different durations and for a more prolonged time than E2, altering the normal responsiveness of prostate cells. However, DBP alone seemed not to influence AR localization, but AR was continuously and persistently activated when DBP was used in combination. Our results show that DBP alone, and in mixture, alters redox homeostasis in prostate cells, leading to a greater increase in cell oxidative susceptibility. In addition, we also demonstrate that DBP increases the migratory potential of PNT1A cells. In conclusion, our findings demonstrate that DBP, alone and in mixtures with endogenous steroid hormones, acts as an EDC, resulting in an altered prostate cell physiology and making these cells more prone to cancer transformation.

## 1. Introduction

Several studies have revealed that the decline in male reproductive health can be related to the exposure to toxic substances found in the environment, particularly endocrine-disrupting chemicals (EDCs) such as phthalates [1,2,3,4,5]. Phthalates are a family of aromatic chemical compounds mainly used as plasticizers [6]. They are widely used in many types of consumer products, such as cosmetics and personal care products, food packaging, medical equipment, and furniture materials [6,7]. As phthalates are noncovalently bound to these products, they are easily released in the environment, exposing organisms to their uptake through inhalation, ingestion, and dermal absorption [6,8]. Based on their molecular weight, phthalates are grouped into two main classes. Di-n-butyl phthalate (DBP) is a low-molecular-weight phthalate found as a component of several cosmetic products, such as nail polish, where it is used for its plasticizing capacities. In addition, due to its solvent properties, it is also present in other personal care products. Its wide use makes it particularly dangerous for human health [9,10,11].

Most tissues of the human body are able to excrete phthalates within 24 hours, with the exception of the liver and lungs, which retain them for a longer time [9,10,11,12]. Several studies have shown the presence of high concentrations of DBP in human urinary metabolites and seminal plasma, indicating high environmental exposure and, thus, explaining the growing interest in evaluating its effects on human and animal health [13,14,15,16]. Previous studies indicated that phthalates are involved in the progression of hormone-related cancers and other diseases, including breast cancer [17], thyroid cancer [18], endometriosis [19], and ovarian cancer [6,20]. Moreover, it has been shown that DBP has direct toxic effects, not only on the endocrine system, but also on other systems and individual organs, including the liver, spleen, kidneys, cardiovascular system, and nervous system [9,11,21,22,23,24,25]. DBP has also been associated with obesity, allergies, and asthma [11,26], as well as with toxic effects on the reproductive system, as it is believed to be responsible for testicular damage, decreased sperm motility, and developmental malformations of the reproductive system [25,27,28,29,30]. According to Mylchreest et al. [31], the lowest observed adverse effect level (LOAEL) dose of DBP in the male genital system of rats is 100 mg/kg [16,32,33]; in addition, it has been shown that higher doses, of 500–850 mg/kg, induce clearer and more evident effects in animals [16,34,35,36,37]. Concerningly, the amount of DBP in the environment significantly exceeds the safety limits defined by national and international standards in countries like the United States and China [25,38], prompting the restricted use of phthalates in children’s toys in the United States since 2008 [6,39]. Furthermore, restrictions on five types of phthalates—di-2-ethylhexyl phthalate (DEHP), DBP, diisononyl phthalate (DiNP), butyl benzyl phthalate (BBP), and di-isodecyl phthalate (DIDP)—were imposed in Europe starting in 2007 [6,40]. Due to the ability of phthalates to cross the placenta [41], maternal exposure to phthalates can induce different effects on the reproductive system, particularly phthalate syndrome (PS), which is characterized by morphological and physiological dysfunction of the testes, including reduced anogenital distance, cryptorchidism, hypospadias, reduction in germ cells, impaired sperm quality, decreased testosterone levels, and delayed onset of puberty. In addition, PS is also characterized by malformations of the epididymis, vas deferens, seminal gland, and prostate in adulthood [16]. The same reproductive symptoms in human males have been defined as testicular dysgenesis syndrome (TDS) [42].

Several studies have shown that phthalates can bind ERα and modulate its activity [6,11,43], thereby acting as EDCs. It is known that ERα, expressed in stromal cells, induces the expression of growth factors that stimulate the proliferation of epithelial cells and stromal cells [44,45]. Increased activation of ERα also directly promotes the progression of prostate hyperplasia and inflammation [44,46,47]. In human ERα and Erβ transactivation assays, a panel of 22 phthalates were investigated for their estrogenic activities. Nine of these compounds were shown to exhibit an ERα-mediated estrogenic activity, whereas seven showed an antiestrogenic activity [11,43]. More recently, it has been demonstrated that phthalates can act as weak human ERα agonists, but they can also be weak selective estrogen receptor modulators (SERMs). These findings suggest that these compounds can act as either agonists or antagonists of ERs in a tissue-specific manner [11,48,49]. In addition, nine phthalates were shown to possess weak antiandrogenic activities by a human androgen receptor (AR) transactivation assay [11,43].

Inside the male reproductive system, sex steroids control the prostate gland, and their regulation is important for the gland’s development and function [44,50]. For this reason, the prostate gland represents an important target of EDCs. In the prostate epithelial and stromal cells, androgens, including testosterone (T) and dihydrotestosterone (DHT), can promote cell proliferation by interacting directly with the ARs [44,45,51]. It has been demonstrated that androgen levels decrease with age, differently to 17β-estradiol (E2): While androgen levels decrease with age, this does not apply to 17β-estradiol, which determines an increase in the E2:T ratio [44]. EDCs interfere with steroid hormone pathways and can increase cell proliferation, thus leading to benign prostatic hyperplasia (BPH) and prostate cancer (PCa) [44]. Zhou et al. [52] demonstrated that DEHP increased E2 levels and decreased androgen levels in male adult rats. These results suggest that DEHP-induced fluctuations of estrogen and androgen levels might underlie the predisposition of prostatic enlargement. In industrialized countries, benign and malignant prostate disorders are among the most common diseases in men [53,54]. Worldwide, PCa represents the second-most common solid tumor in men [25,55], being the second-most prevalent global cancer in 2020, with the sixth highest cancer mortality rate, and accounting for up to 7.3% of all global cancer incidence [6,56]. As another kind of cancer, prostate cancer is characterized by multiple biological processes including several genetic and epigenetic alterations that affect epithelial cells, considered the main target cells. It has been demonstrated that the long and complex process of prostate carcinogenesis can be also influenced by lifestyle factors such as diet, alcoholism, smoking, and obesity [37,54,57,58]. It is widely recognized that exposure to chemical compounds such as phthalates and other EDCs plays an important role in the development of PCa [25,59,60], which is considered a hormone-dependent tumor [6,61]. In experimental studies, phthalates and their metabolites stimulate cell proliferation in PCa by upregulating the expression of AR [44,57], ERα [10,44,62], and specific genes (*ERK5, p38, KLK3*, and *PTCH*) [44,63,64]. Moreover, Zhou and colleagues [52] showed that DEHP significantly increased the ratio of PCNA/TUNEL, suggesting a role in the onset of benign prostatic hyperplasia [52]. Several studies show that estrogen and androgen signaling pathways perform physiological functions by interacting directly with their cognate receptors [50,52,65]. Previously, Zhang et al. demonstrated that an increased cell proliferation was induced in LNCaP by very low concentrations of DBP, as compared to the control group [66]. Moreover, due to the significant role of the androgen receptor in the occurrence and development of PCa [66,67], the same study also demonstrated that DBP and its metabolites are able to bind to AR similarly to testosterone, suggesting an androgen-blocking effect of DBP [43,66].

Many studies have demonstrated that in vivo and in vitro exposure to phthalates (including DBP and DEHP) causes oxidative stress (OS), primarily by reducing superoxide dismutase (SOD) and glutathione (GSH) levels [44,67,68,69], and by causing OS-related disorders [44,70]. Specifically, it has been demonstrated that DEHP and its metabolite interfere with the redox activity at the molecular level, reducing the activity of the crucial mammalian antioxidant enzyme, glutathione peroxidase 1 (GPx1) [6]. It has been shown that by decreasing antioxidant capacity, phthalates have a devastating effect on DNA structure in LNCaP cells [6,71]. Recently, Thomas et al. [72] demonstrated that MEHP and PFOS are able to perturb about five thousand proteins involved in biological processes such as PPAR signaling (e.g., lipid metabolism and oxidative stress), proteostasis, and damage-associated molecular pattern (DAMP) signaling, ultimately leading to innate immune response signaling, which is critical to both BPH and PCa [72]. However, the underlying mechanism through which phthalates affect the prostate gland remains unclear. The aim of this research was to investigate the effects of DBP, alone and in combination with testosterone, estradiol, and both, on the normal human prostate-derived PNT1A cell line, to unravel how these substances combine with other endocrine disruptors and natural hormones. Our results show that DBP interferes with cell viability, steroid receptor levels and localization, the redox homeostasis, and migratory potential in PNT1A cells.

## 2. Results

### 2.1. DBP Increases Cell Viability of Prostate Cells

We investigated the effects of DBP and endogenous sex hormones T and E2 on cell viability by MTT assay. Our results showed that all of these compounds, when used alone, were able to increase cell viability, although different concentrations were required. In particular, testosterone increased cell viability at 10^–8^ M (Figure 1A), whereas E2 was at 10^–6^ M (Figure 1B); DBP was able to increase PNT1A viability when used at 10^–9^ M, indicating that the concentration of this compound required to affect cell viability is significantly lower than that required by sex hormones (Figure 1C). When used in combination, all mixtures determined an increase in cell viability, the greatest effects being observed when using T alone and with the mixture DBP + T (Figure 1D). Conversely, despite the mixture of DBP + T + E2 showing an increase in cell viability, it was slightly lower than the effect elicited by hormones alone or coupled with DBP, suggesting a possible competitive mechanism (Figure 1D). To show the possible involvement of steroid receptors, all of the experiments were performed also using estrogen and androgen receptors inhibitors, ICI_182,780_ and flutamide, respectively. ICI_182,780_ was strongly able to decrease the effects of all the experimental classes containing E2 (alone or in combination with DBP), whereas it slightly influenced the increasing effect of DBP and T alone; on the contrary, ICI_182,780_ strongly reduced the proliferative effects of mixed DBP + T (Figure 1E). Except for E2, and to a lesser extent for DBP + E2, flutamide decreased cell viability in all the experimental classes, which showed a viability comparable to the control (Figure 1F).

### 2.2. DBP Alters ERα and AR Protein Expression

To understand whether steroid receptors were affected by exposure to DBP, alone and in combination with endogenous hormones, protein levels of ERα and AR were detected in PNT1A cells following treatment for 30 min, 2 h, and 4 h.

The result observed depended on the duration of the exposure: when cells were treated for 30 minutes with DBP, ERα level did not change; however, after 2 h and 4 h treatment, DBP was able to strongly decrease ERα levels (Figure 2A,B). E2 also decreased ERα level, showing an effect already after 30 min (Figure 2A,B). On the contrary, T was able to decrease ERα expression after 30 min and even more after 2 h, but the expression of ERα resembled that in the control group after 4 h (Figure 2A,B). When using mixtures, the effects depended on the combinations; DBP + E2 decreased ERα levels after 30 min and again after 4 h, but induced an increase after 2 h, showing a different modulation compared to DBP or E2 used alone (Figure 2A,B). Conversely, DBP + T was able to decrease ERα amount after 30 min and 2 h similarly to T, but in a more pronounced way; differently to T alone, DBP + T induced an increase in Erα levels after 4 h (Figure 2A,B). The mixture T + E showed a mode of action similar to E2, as it induced a decrease in Erα level in all the experimental treatment times (Figure 2A,B). Finally, the mixture DBP + T + E2 showed a nonlinear effect, as it decreased ERα levels after 30 min, but it induced a strong increase in ERα levels after 2 h, again inducing a slight decrease in ERα after 4 h (Figure 2A,B).

When measuring AR levels, DBP was able to strongly increase AR expression after 30 min and after 2 h, showing a less pronounced effect after 4 h (Figure 2A,B). The opposite behavior was instead shown for E2, which strongly decreased AR levels at all treatment times (Figure 2A,B). Our results also showed that T decreased AR amount after 2 h, but then strongly increased AR levels after 4 h (Figure 2A,B). When used in combination, DBP + E2 had a similar effect to that of E2 alone, inducing a decrease in AR levels after 30 min and 2 h. However, after a 4 h treatment, the mixture DBP + E2 strongly increased AR, showing a different effect than DBP and E2 alone (Figure 2A,B). On the contrary, the mixture DBP + T induced a strong decrease in AR levels after 30 min and 2 h treatment, but strongly increased AR amount after 4 h (Figure 2A,B). Moreover, the mixture T + E reduced AR levels after 30 min and 2 h, but increased AR amount after 4 h, even if the effect was less pronounced than that induced by T (Figure 2A,B). Finally, the mixture DBP + T + E2 had a nonlinear behavior, strongly decreasing AR after 30 min, resembling the control after 2 h, increasing its expression after a 4 hour treatment, but to a lesser extent than the mixture DBP + E2 and DBP + T (Figure 2A,B).

### 2.3. DBP Treatment Affects ERα Cellular Localization

To investigate whether DBP could activate the ERα pathway, the cellular localization of ERα was observed by immunofluorescence after a 30 min, 2 h, and 4 h DBP treatment (Figure 3, Figure 4, Figure 5, Figure 6, Figure 7 and Figure 8). Our results showed that a 30 min treatment did not affect ERα localization, as it can be observed in the cytoplasm (Figure 3). However, DBP induced ERα nuclear translocation after 2 h and 4 h (Figure 3). In contrast, E2, the natural ERα ligand, was able to induce an earlier nuclear translocation of ERα, which could be observed in the nucleus after 30 min, and the nuclear localization also persisted after 2 h. However, ERα could be observed back to the cytoplasm after 4 h (Figure 4). The mixture DBP + E2 showed a similar effect to E2 treatment alone, as it induced nuclear translocation after 30 min and 2 h (Figure 5). Differently, the mixture DBP + T did not translocate ERα to the nucleus in any of the time course (Figure 6). The mixture T + E2 induced nuclear translocation after 2 h and after 4 h (Figure 7); a similar result was observed following treatment with the mixture DBP + T + E2 (Figure 8).

### 2.4. DBP Treatment Does Not Affect AR Cellular Localization

The effect of DBP on androgen pathway was investigated by monitoring AR localization by immunofluorescence (Figure 9, Figure 10, Figure 11, Figure 12, Figure 13 and Figure 14). DBP treatment did not induce AR translocation to the nucleus in any of the treatments (Figure 9). On the contrary, T was able to slightly translocate AR to the nucleus after 30 min and 4 h treatments, showing a cyclic behavior inside the cell (Figure 10). We then checked the effect of the mixture DBP + T, which showed a similar trend on AR translocation, as a nuclear localization was observed after 30 min and again after 4 h (Figure 11). In contrast, the mixture DBP + E2 did not induce a nuclear translocation of AR, even if a perinuclear localization of AR could be observed in all the experimental times (Figure 12). The mixture T + E2 consistently induced AR nuclear translocation after 30 min, 2 h, and 4 h (Figure 13). Similarly, the mixture DBP + T + E2 strongly and constantly translocated AR to the nucleus in all the experimental times (Figure 14).

### 2.5. DBP Exposure Triggers Oxidative Stress

To evaluate whether DBP treatment could induce oxidative stress in PNT1A cells, the redox-homeostasis-related parameters were measured. DBP alone induced a slight but significant increase in lipid hydroperoxides levels (Figure 15A), which are an index of lipid oxidative damage; on the contrary, when treating cells with the mixtures DBP + T and DBP + E2, levels of lipid hydroperoxides remained unchanged. Also, E2 and T alone did not affect the levels of lipid hydroperoxides. Conversely, DBP + T + E2 induced a significantly increase in lipid oxidative damage similar to that obtained by the T + E2 treatment. Our results showed that E2 and DBP significantly increased susceptibility to oxidative stress with respect to the control group (Figure 15B). On the other hand, T alone and T + E2 did not affect this susceptibility. The mixtures DBP + T and DBP + T + E2 were strongly able to increase the oxidative stress susceptibility. Moreover, the total ROS content (Figure 15C) showed that only the mixture DBP + T + E2 significantly increased it. Finally, we monitored the nonenzymatic antioxidants capacity (Figure 15D), which was increased only by the mixtures DBP + T and T + E2 compared to the control group.

### 2.6. DBP Treatment Increases Cell Migration in PNT1A Cells

A cell mobility assay was performed to investigate the effects of DBP on cell migration when used alone and in mixtures (Figure 16). An increased cell migration was observed in all of the experimental classes, except for T alone and the mixture T + E2 (Figure 16A). In particular, DBP and E2 alone were able to increase cell migration, showing an even greater effect on cell velocity when used in combination. Moreover, whereas T did not affect cell mobility, a strongly increase in cell migration was observed when used in combination with DBP, in the mixture DBP + T. Conversely, the mixture DBP + T + E2 did not show any effect on cell migration. To investigate a possible involvement of steroid receptors in cell migration, wound healing assay was also performed after treatment with ERα and AR inhibitors (Figure 16B,C). DBP and E2 alone did not affect cell migration if used after ICI_182,780_ treatment (Figure 16B). In all of the other experimental classes containing E2, an increase in cell migration was observed, particularly in the mixture of DBP + T + E2 (Figure 16B). Moreover, when ERα was inhibited, an increase in cell migration was observed after T treatment (Figure 16B). In addition, we performed scratch assay after flutamide treatment, which showed that, with the exception of the mixture DBP + T, all of the experimental classes were able to increase cell migration, with a peak after E2 and T + E2 treatments (Figure 16C).

## 3. Discussion

The prostate gland is a sex-steroid gland under the control of androgen and estrogen hormones, whose balance is important to maintain gland homeostasis and physiological conditions. It has been demonstrated that androgen/estrogen imbalance during aging could stimulate prostate cell proliferation, which is responsible for several prostate pathologies [45,73]. Simultaneously, the balance of cell proliferation and cell death can be impaired under the cooperation between growth factors and steroid hormones, which can onset BPH [48,73]. Sex hormones can interact with nuclear steroid receptors, namely, estrogen and androgen receptors. Specifically, in the prostate gland, the classic estrogen receptor, ERα, is primarily located in the stroma and basal cell layers that contain proliferative stem cells [6,74]. It has been shown that ERα can function as an oncogene, as revealed by numerous molecular and epidemiological studies [6,75]. Previous findings suggested that direct estrogenic activity mediated by estrogen receptors regulates the progression of PCa [6,76]. In light of this evidence, substances affecting the estrogen and androgen signaling pathways, such as EDCs, can potentially impact prostate homeostasis, thus inducing several pathologies, and culminate in PCa development and progression [11]. However, despite decades of scientific research, the relevance of EDCs exposure on prostate biology is still lacking [11]. Phthalates are a group of compounds belonging to EDCs widely used in many fields to improve the quality of plastics; for this reason, they become the most identified environmental chemicals [66,77]. Despite its largest production cost, the phthalate DBP is one of the most used plasticizers [66,77]. Plasticizers are added to crude plastic products through mechanical mixing rather than covalent bonding, which makes them easily released into the environment, in food, and in the drinking water, due to vibration or heating [66,78]. Phthalates have been detected in human blood, milk, and urine [66,79,80]. Several studies have identified a direct relationship between the exposure to phthalates and pathologies of different organs and systems such as metabolic disorders, damage to the reproductive system, and hormone-sensitive cancers [54,80,81], including breast cancer [66,82]. However, the relationship between phthalate and prostate gland pathologies needs further investigation [66].

Many EDCs, particularly bisphenols, phthalates, phytoestrogens, and mycoestrogens, are molecules that interact with the androgen and estrogen signaling pathways in diverse ways. Their direct binding to ERs, AR, or both has been described, or they can have an indirect effect by impacting steroidogenesis, expression of sex-steroid receptors via epigenetics, or the activation mechanisms of the nuclear receptors [11]. However, many of these chemicals can act on multiple cellular targets, depending on their concentration or cell cycle phase, making it more difficult to understand their mechanisms of action [11].

In this study, we evaluated the effects of DBP alone and in combination with endogenous sex hormones, testosterone, estradiol, or both on the normal human prostate-derived PNT1A cell line, to mimic the environmental contamination. We aimed at understanding the possible interaction among DBP and endogenous male and female hormones in order to highlight agonistic, antagonistic, or competitive mechanisms. PNT1A cells were chosen as the experimental model because both estrogen and androgen receptors are expressed, making them an ideal model for evaluating the effects of these substances on sex hormones signaling pathways. Even though the use of a single cell line can represent a limitation of the study, experiments were carried out on a single cell line to better unravel cell-specific mechanisms of action. Our results showed that the DBP, used alone and in all of the tested mixtures, can increase prostate cell viability. We showed that DBP strongly increased cell viability at a concentration lower than T and E2 when treating cells for 24 h. When used in mixtures, DBP maintained proliferative effects, even if competitive effects seemed to be activated. Using specific nuclear steroid receptor antagonists, we showed that the increase in proliferative effect is probably mediated by activation of both ERα and AR, as their nuclear translocation was induced. It has been previously shown that phthalates and their metabolites act on prostate cell proliferation in PCa through affecting the production of AR and ERα, and interfering with gene pathways such as P38 [10,57,63,73]. Moreover, it has been demonstrated that specific phthalates, including DEHP, promote cell proliferation by activating the MAPK/AP-1 pathway [6,65] and increasing expression of *KI67* in PC-3 and 22RV1 PCa cells [54,83].

As suggested by Lacouture et al. [11], to deeply understand the mode of action of EDCs, it is fundamental to estimate the expression of sex-steroid receptors such as AR and ERα, to evaluate whether the effects resulting from EDC treatments are mediated by these hormones [11]. In addition, positive and negative controls (agonists and antagonists) of these receptors must constantly be evaluated in parallel to EDCs [11]. E2, the most biologically active estrogen, can be used as a positive control for ERs because of its strong affinity for both receptors [11]. Furthermore, pure antiestrogens, such as ICI_182,780_, should be used to demonstrate if the effects observed after EDC treatment are induced by ER activation [11,84,85,86,87]. Similarly to ERs, antiandrogens, such as flutamide, should also be used to block and inhibit AR-dependent activation [11]. In our study, we assessed the interference of DBP on steroid pathways and we showed that this phthalate was able to perturb the physiological conditions of human prostate cells PNT1A by interfering with nuclear steroid expression, and inducing a possible dysregulation of cell functions. Furthermore, we showed that DBP, alone and in combination with T and/or E2, was able to modulate levels of both ERα and AR in a nonmonotonic way, and differently to endogenous hormones. Moreover, we showed that DBP was able to translocate ERα to the nucleus and for a more prolonged duration than E2, thus altering the normal responsiveness of prostate cells. On the contrary, DBP alone seemed to not influence AR localization, but when used in the mixture with both T and E2, DBP maintained AR in the nucleus, suggesting a continuous and persistent activation of this receptor. For the first time, our results showed that the simultaneous presence of DBP and endogenous hormones can strongly affect prostate cell homeostasis. Due to the constant and wide presence of DBP in the environment, our results may be used as a red flag to prevent and control environmental contamination from EDCs.

Previous studies have demonstrated that phthalates can interfere with redox activity and antioxidant capacity of cells [72]. In the present study, therefore, we evaluated both the effect DBP exposition alone and in mixture with T and E2 on redox homeostasis. We showed that DBP alone, and in mixture with both T and E2, analogously to T + E2 mixture, significantly increased the hydroperoxide content compared with the other groups. This increase could be due to the ability of DBP to induce oxidative damage, even if this ability is also dependent on the exposure time [24]. However, the total reactive oxygen species content was not affected, despite an increased oxidative damage. We observed that a significant increase in ROS content only occurs in cells treated with the mixture of DBP, T, and E. In this group, ROS content increase does not lead to an increase in the nonenzymatic antioxidant defenses, evaluated with the ABTS dosage. A significant increase in this ability was only observed in the group treated with T and E2, compared with the control group, and the group treated with T alone. These cells appear to be less susceptible to induction of oxidative stress conducted in vitro. On the other hand, in the group treated with the three-way mixture, in the absence of an increase in antioxidant defenses, there is a greater increase in in vitro susceptibility as compared to all the other groups. These data are in line with the evidence that exposure to hormonal imbalance and phthalates can induce the onset of oxidative stress [26]. In a previous work, Cavalca et al. [54] showed that an environmentally relevant mixture of phthalates affected the antioxidant system by inducing an increase in expression of important cellular antioxidants, such as CAT, GSR, and SOD1, in LNCaP cells at 24 h, suggesting a probable increase in cellular oxidative stress [54,88,89,90]. The dysregulation of the antioxidant system was also showed by Ferguson [88] through analyses that quantified the presence of phthalate metabolites in the urine of 10,026 individuals [54]. It has been demonstrated that the imbalance in the production and detoxification of OS can be associated with the early stage of prostatitis and the development of BPH [70,73]. Yang and collaborators found that DEHP may induce oxidative stress, which causes inflammation and, hence, results in BPH [44,73]. In vivo, DEHP is rapidly metabolized, and its metabolites can destroy mitochondrial function and produce excessive reactive oxygen species. Increased oxidative stress levels can generate cell injury and inflammation by damaging DNA and lipids [44,73,91,92]. Moreover, it has been shown that low-dose DEHP induces an unbalanced E2/T ratio and increased COX-2 and L-PGDS expressions that might be responsible for developing prostatic hyperplasia [52]. Since androgen, estrogen, and ROS are sensitive to being exposed to phthalates, changes in sex hormones and in ROS might be critical for comediating prostate enlargement in elderly men [26,44,68].

It is known that different phthalates have different actions [54,93]. Specifically, it has been demonstrated that phthalates can increase migration and invasion, negatively regulating genes responsible for tumor suppression, as observed by Huang et al. [54,94,95,96,97], validating studies in the literature that showed the interference of phthalates on cell motility and cancer progression [54,97]. Our results showed that DBP alone and in mixture with T or E2 was able to increase the migratory potential of human prostate cells. Moreover, we observed the role of both nuclear steroid receptors in the migration process, showing that their inhibition can promote migratory potential of PNT1A. Intriguingly, we cannot exclude the involvement of the G protein-coupled estrogen receptor (GPER or GPR30) located at the plasma membrane.

In conclusion, the present work highlights the molecular mechanisms activated by DBP alone and in combination with endogenous hormones, both male and female. Many EDCs can act on multiple cellular targets depending on their concentration or cell cycle phase, making it more difficult to understand their mechanisms of action. In vitro data are essential to identify the main effect markers to be used in studies on EDCs. Our results represent an important starting point for highlighting the cell-specific mechanisms of action, to better tailor the experimental design by the use of in vivo studies.

Our experimental work demonstrates that DBP alone and in mixture with testosterone, estradiol, or both increases cell viability by inducing a modulation of ERα and AR protein levels and their subsequent activation. In addition, we showed that DBP is able to increase oxidative stress and the migratory potential of normal PNT1A prostate cells, thus suggesting that DBP, acting as EDC, increases susceptibility of these cells to malignant transformation.

## 4. Materials and Methods

### 4.1. Chemicals

Testosterone (T), 17β-estradiol (E2), dibutylphthalate (DBP), flutamide (FLUT), and fulvestrant (ICI_182,780_) were purchased from Sigma-Aldrich (Sigma Aldrich, St. Louis, MO, United States) and dissolved in DMSO (Sigma Aldrich, St. Louis, MO, United States). T, E2, and DBP were diluted in RPMI 1640 red-phenol free (Sigma Aldrich, St. Louis, MO, United States), reaching final concentrations from 10^−6^ to 10^−12^ M, and mixed to create the experimental class used in this study. Instead, ICI_182,780_ and FLUT were diluted in RPMI 1640 red-phenol free at final concentrations of 10^−5^ M and 10^−7^ M, respectively.

### 4.2. Experimental Design

For each assay, cells were seeded according to previous calibrations, starved for 24 h, and treated with chemicals. ICI_182,780_ and FLUT were added 1 h prior to mixture treatments to block their specific receptors. For MTT and wound healing assay, the cells were treated for 24 h; instead, for immunofluorescence assays and Western blotting (WB), the cells were treated for 30 min, 2 h, and 4 h, times necessary to verify if the cytoplasm–nucleus translocation takes place.

### 4.3. Cell Culture

The prostate nontumoral cell line established by immortalization of adult prostate epithelial cells (PNT1A cells, ECACC 95012614) was maintained in RPMI medium (Sigma Aldrich, St. Louis, MO, United States), supplemented with 10% fetal bovine serum (FBS, Sigma Aldrich, St. Louis, MO, United States), 2 mM L-glutamine (Sigma Aldrich, St. Louis, MO, United States), and 100 U/mL penicillin/streptomycin (Sigma Aldrich, St. Louis, MO, United States) in a humidified incubator at 37 °C and 5% CO_2_ [98]. After 70% confluency, cells were harvested with Trypsin/EDTA solution (Sigma Aldrich, St. Louis, MO, United States) and cultured into new flasks. The culture medium was replaced twice a week. Cells were monitored daily using an inverted microscope.

### 4.4. MTT Assay

MTT assay was performed to establish the concentration of T, E2, and DBP to be used in mixtures for cell treatments. The MTT test is a colorimetric assay that is used to evaluate cellular metabolic activity, allowing the assessment of the effects of different doses of chemicals on cell viability, proliferation, and cytotoxicity [53]. The term “cell viability” is the term employed to indicate a variety of cell markers used to evaluate the healthy cell within a culture (example: metabolic activity, ATP content). The aim of cell viability assay is to measure cell survival or cell proliferation following treatment with a specific compound, evaluating its toxicity. To sum up, cell viability is a useful tool often used in in vitro experiments to evaluate the cellular response after a specific drug treatment. Then, the MTT assay was carried out to evaluate the potential antagonist or synergistic effects of the mixtures mentioned above. PNT1A cells were seeded in a 96-multiwell plate at a density of 5000 cell/well. After 24 h of treatment, 10 μL of MTT solution (5 mg/mL) was added to each well and incubated in a humified incubator for 3 h [99]. The formazan crystals created in each well were dissolved in DMSO. The absorbance at 570 nm was read by microplate reader. The cell viability (%) was obtained as OD570 sample–blank/OD570 control–blank × 100. MTT assay was performed in triplicate.

### 4.5. Total Protein Extraction and Western Blotting

After 30 min, 2 h, and 4 h, cells were harvested using a scraper and subjected to total protein extraction using the RIPA buffer (Tris 50 mM, NaCl 150 mM, SDS 0.1%, Na Deoxycholate 0.5%, NaF 5 mM, NP40 1%, EDTA 10 mM) enriched with protease-inhibitor cocktail (Sigma Aldrich, St. Louis, MO, United States). Protein concentration was measured with BCA protein assay (Thermo Fisher Scientific, Waltham, MA, United States), according to the manufacturing protocol. Then, 40 μg of protein were boiled for 5 min in SDS buffer (50 mM Tris-HCl (pH 6.8), 2 g 100 mL^−1^ SDS, 10% (*v/v*) glycerol, 0.1 g 100 mL^−1^ Bromophenolblue), separated on 10% SDS-PAGE and transferred to a PVDF membrane for blotting (Trans-Blot^®^ Semi-Dry Transfer Cell, Bio-Rad Laboratories, Hercules, CA, United States). Membranes were then stained with ponceau red to verify the successful transfer. After washing, they were incubated for 1 h with blocking buffer (TBS, 0.05% Tween-20 and 5% bovine serum albumin, BSA) at room temperature and then incubated with primary antibodies diluted in TBS-T containing 2% BSA overnight at 4 °C. Primary antibodies used were ERα Santacruz, Dallas, TX, United States, sc8005, mouse monoclonal, 1:200) and AR (Abcam, Cambridge, United Kingdom, Ab74272, rabbit polyclonal, 1:200). Membranes were washed four times for 10 min in TBS, 0.05% Tween-20, before a 1 h incubation with secondary antibody diluted in TBS-T containing 2% BSA. Secondary antibodies used were goat anti-rabbit IgG (HRP) (Abcam, Cambridge, United Kingdom, Ab6721, 1:3000) and goat anti-mouse IgG (Santacruz, Dallas, TX, United States, sc-2354, 1:3000) [98,99]. Then, membranes were washed four times for 10 min and specific protein bands were detected with chemiluminescence using a ChemiDoc Image System (Bio-Rad Laboratories, Hercules, CA, United States). Western blot results were analyzed using ImageJ 1.54F to determine optical density (OD) of the bands. The OD reading was normalized to GAPDH to account for variations in loading.

### 4.6. Immunofluorescence

PNT1A cells were seeded at a density of 2.5 × 10^4^/well in 8-well chamber slides (Sarstedt, Nürnbrecht, Germany). After three different times of treatments (30 min, 2 h, and 4 h), cells were fixed with ice-cold methanol for 5 min at RT, washed in PBS, and blocked in 1% bovine serum albumin (BSA) in PBS–Tween20 for 30 min. Afterwards, cells were incubated with the following primary antibodies overnight at 4 °C: ERα (Santacruz, Dallas, TX, United States, sc8005, mouse monoclonal, 1:100) and AR (Abcam, Cambridge, United Kingdom, Ab74,272, rabbit polyclonal, 1:300). The following day, cells were incubated with secondary antibody diluted in PBS. Secondary antibodies were Goat anti-Mouse IgG (H&L) DyLight^®^ 488 Conjugate (Immunoreagent Raleigh, North Carolina NC, United States, GtxMu-003-D488NHSX, 1:250) and Goat anti-Rabbit IgG (H&L) DyLight^®^ 594 Conjugate (Immunoreagent Raleigh, North Carolina NC, United States, GtxRb-003-D594NHSX, 1:250). Nuclei were stained with Höechst (Thermo Fisher Scientific, Waltham, MA, United States) [98]. All morphological observations were carried out using a Zeiss Axioskop microscope, and images were acquired by using an Axiovision 4.7 Software (Zeiss, Oberkochen, Germany) epifluorescence microscope. 

### 4.7. Redox Homeostasis

Lipid hydroperoxides (HPs), markers of lipids oxidative damage, were assayed according to Heat and Tappel [100] following the reduction in the NADPH concentration (λ = 340 nm) in a system of two coupled reactions. In the first reaction, the enzyme glutathione peroxidase reduced the HPs of the sample (10 µg of cell proteins in 0.1 M monobasic phosphate buffer, pH 7.4) by oxidizing glutathione. The oxidized glutathione was reduced in the second reaction by the enzyme glutathione reductase that oxidizes NADPH. The content of hydroperoxides levels was expressed as µmol NADPH oxidized∙min^−1^∙mg^−1^ protein. To assess cells’ susceptibility to oxidative stress, the difference between cells’ baseline HPs levels and after incubation with iron and ascorbate to induce oxidative stress was determined [101]. Oxidative stress was induced by incubating cells with a mix of iron and ascorbate (Fe/As, 100/1000 μM) for 10 min at room temperature, and the reaction was stopped by adding 0.2% 2,6-di-t-butyl-p-cresol (BHT). Cellular content of ROS was evaluated according to Fasciolo et al. [102] by determining the ROS-induced oxidation of the nonfluorescent molecule, 2′,7′-dichlorodihydrofluorescein diacetate (DCFH-DA), into a fluorescent molecule, dichlorofluorescein (DCF). The assay has two steps: In the first, 12.5 µg of cellular proteins are incubated for 15 min with 10 µM DCFH-DA in 0.1 M dihydrogen phosphate buffer, pH 7.4. In the second phase, 100 µM of FeCl_3_ is added to the samples and incubated for 30 min. Fluorescence was measured in a multimode microplate reader (Synergy™ HTX Multimode Microplate Reader, BioTek, Winooski, VT, United States). Total antioxidant capacity (ABTS) of cells was detected following the decolorization of the radical 2,2′-azinobis-(3-ethylbenzothiazoline-6-sulfonic acid) (ABTS^•+^) [103,104]. In brief, nonradical ABTS was incubated overnight with potassium persulfate (245 mM) to obtain the radical ABTS^•+^ whose decolorization induced by cells’ antioxidants was then determined at 734 nm. A stock solution of 3,5-Di-tert-4-butylhydroxytoluene (BHT) was used to obtain a calibration curve. Total antioxidant capacity was expressed as BHT equivalents × mg^−1^ protein.

### 4.8. Wound Healing Assay

PNT1A cells were cultured in a 24-multiwell plate until a monolayer formed. The wound was created using a sterile P1000 tip; after washing with PBS (Sigma Aldrich, St. Louis, MO, United States), cells were treated with the mixtures for 24 h. The assay was performed using July stage, which allowed us to follow the scratch in timelapse. Image analysis was carried out through the ImageJ tool, a so-called wound healing size tool. Average rate of healing after 24 h of treatment was measured as the ratio between space and time.

### 4.9. Statistical Analysis

The datasets are expressed as means ± SEM. The statistical analysis was performed using the software Graphpad Prism, version 8.0.2 (GraphPad Software, Boston, MA, United States). Significance was calculated by ANOVA test with Dunnet multiple comparison test, and differences were considered statistically significant when *p* value was at least *p* < 0.05. All experiments were carried out in triplicate.

## Figures and Tables

**Figure 1 ijms-24-14341-f001:**
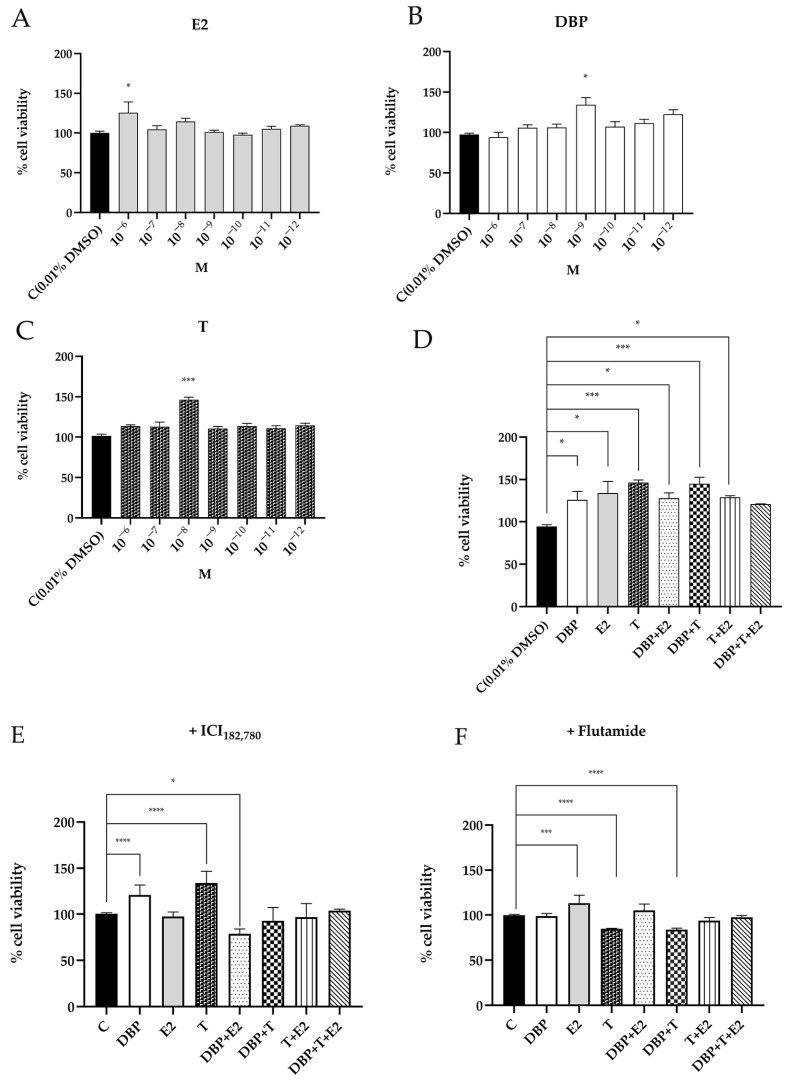
MTT assay after 24 h of treatment with testosterone (T) (**A**), 17β-estradiol (E2) (**B**), and dibutylphthalate (DBP) (**C**). Cell proliferation was induced by E2, T, and DBP at 10^−6^, 10^−8^, and 10^−9^ M, respectively. When used in combination, all of the mixtures stimulated cell proliferation, with DBP + T showing the greatest effect. DBP + E2 + T showed a less proliferative effect than DBP (**D**). When ICI_182,780_ 10^−5^ M was used, the effects were reverted (**E**). Flutamide 10^−7^ M decreased cell viability in all of the experimental classes, except for E2 and, to a lesser extent, DBP + E2 (**F**) (Dunnet’s test: * *p* < 0.05, *** *p* < 0.001, **** *p* < 0.0001).

**Figure 2 ijms-24-14341-f002:**
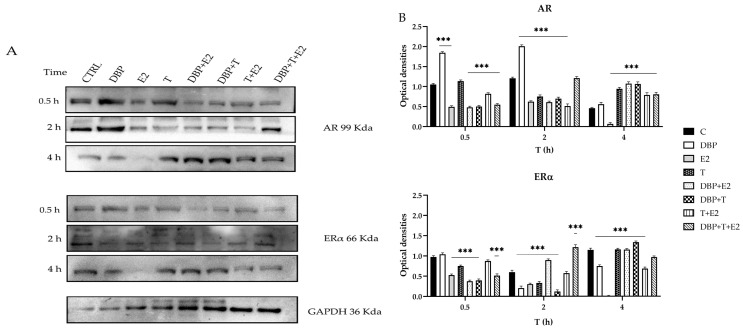
Western blot analysis: quantification of ERα and AR after 30 min, 2 h, and 4 h of exposure to T, E2, and DBP, alone and in mixture. Details provided in the text (**A**). The graphs represent the optical density (OD) ratio of ERα, and AR is normalized to the OD of GAPDH (**B**) (Dunnett test: *** *p* < 0.001).

**Figure 3 ijms-24-14341-f003:**
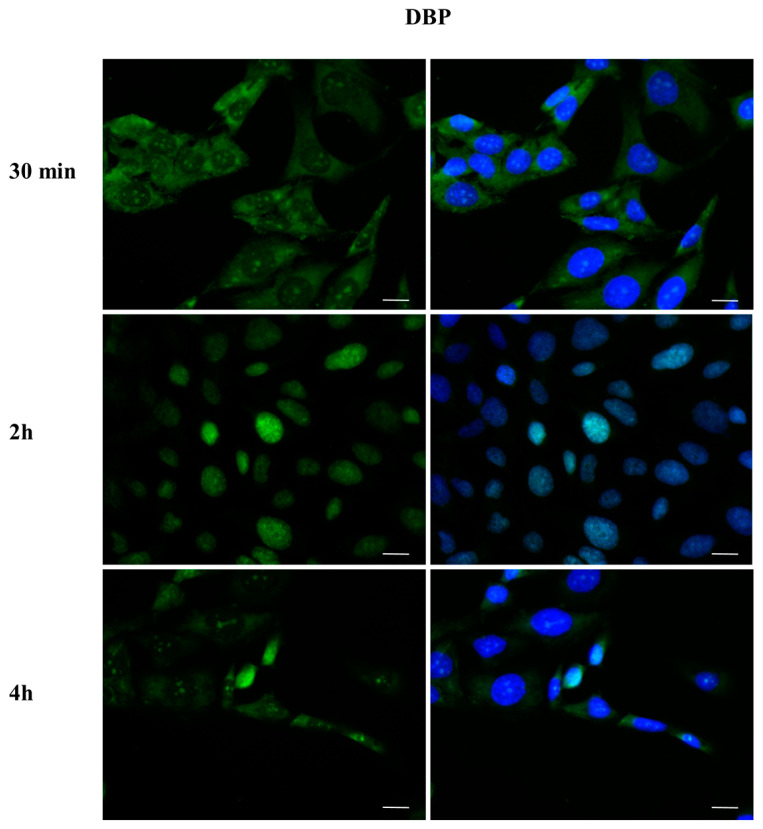
Localization of ERα after 30 min, 2 h, and 4 h of exposure to DBP 10^−9^ M. DBP showed an estrogen-like behavior, inducing ERα cytoplasm–nucleus translocation after 2 h and 4 h. Alexa Fluor 488 (green) and nuclear staining (Höechst-blue) were analyzed by immunofluorescence. Scale bar: 10 µm.

**Figure 4 ijms-24-14341-f004:**
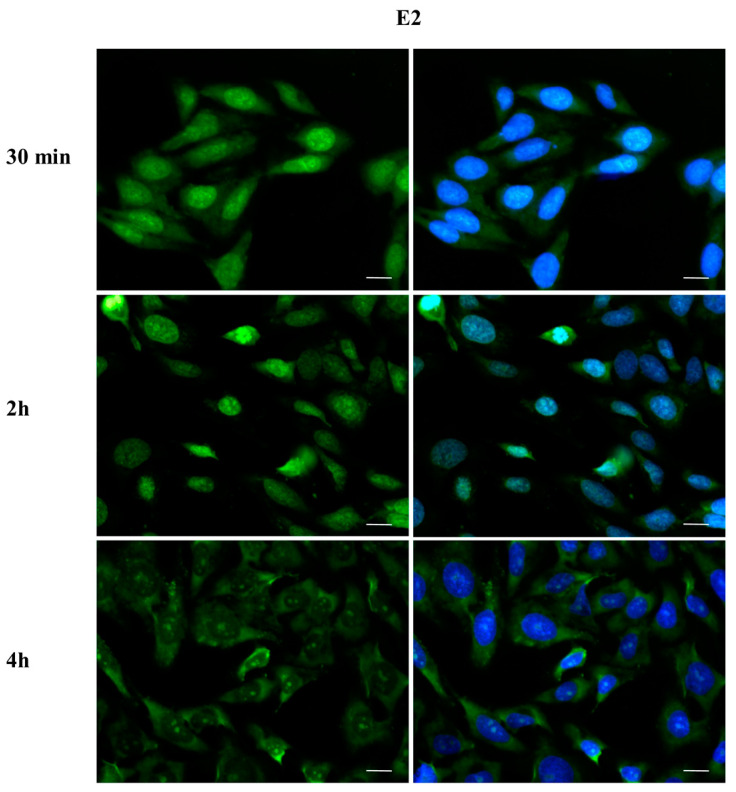
Localization of ERα after 30 min, 2 h, and 4 h of exposure to E2 10^−6^ M. E2 induced cytoplasm–nuclear ERα translocation after 30 min and 2 h. Alexa Fluor 488 (green) and nuclear staining (Höechst-blue) were analyzed by immunofluorescence. Scale bar: 10 µm.

**Figure 5 ijms-24-14341-f005:**
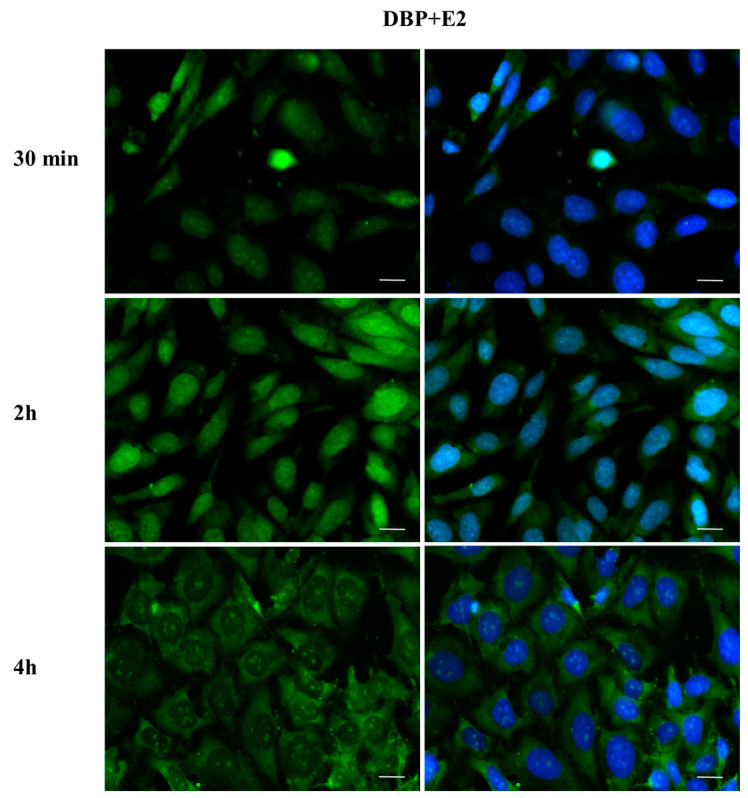
Localization of ERα after 30 min, 2 h, and 4 h of exposure to DBP + E2. The mixture induced cytoplasm–nuclear ERα translocation in the same way as E2. Alexa Fluor 488 (green) and nuclear staining (Höechst-blue) were analyzed by immunofluorescence. Scale bar: 10 μm.

**Figure 6 ijms-24-14341-f006:**
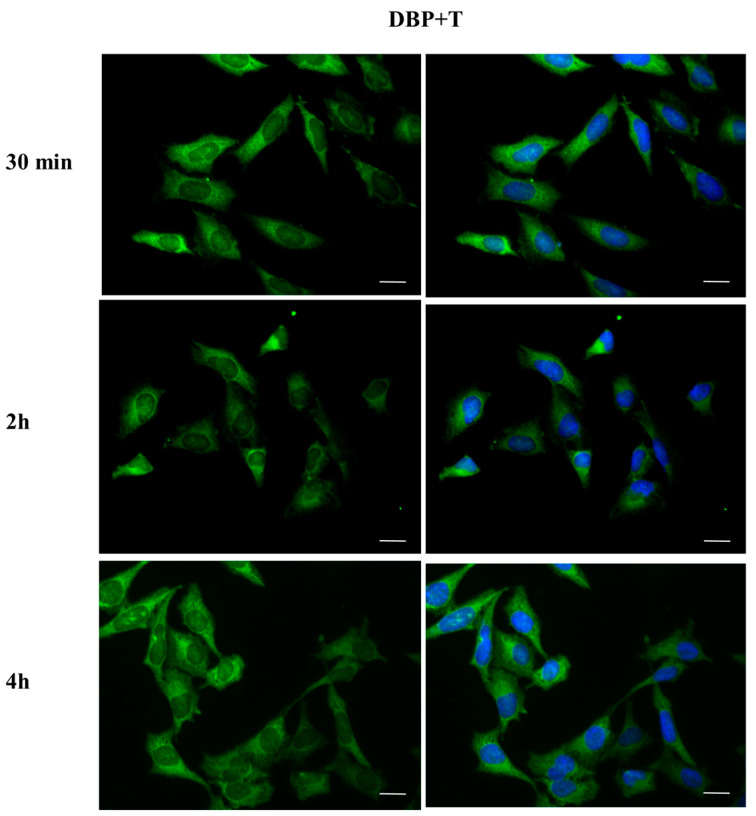
Localization of ERα after 30 min, 2 h, and 4 h of exposure to DBP + T. The mixture did not translocate ERα to the nucleus in any of the treatments. Alexa Fluor 488 (green) and nuclear staining (Höechst-blue) were analyzed by immunofluorescence. Scale bar: 10 μm.

**Figure 7 ijms-24-14341-f007:**
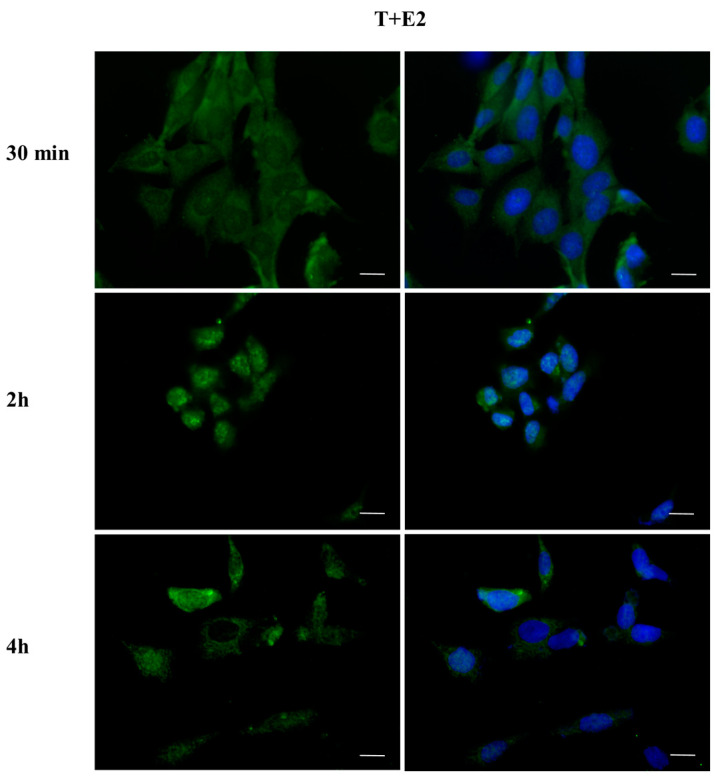
Localization of ERα after 30 min, 2 h, and 4 h of exposure to T + E2. The mixture induced cytoplasm–nuclear ERα translocation after 2 h and 4 h. Alexa Fluor 488 (green) and nuclear staining (Höechst-blue) were analyzed by immunofluorescence. Scale bar: 10 μm.

**Figure 8 ijms-24-14341-f008:**
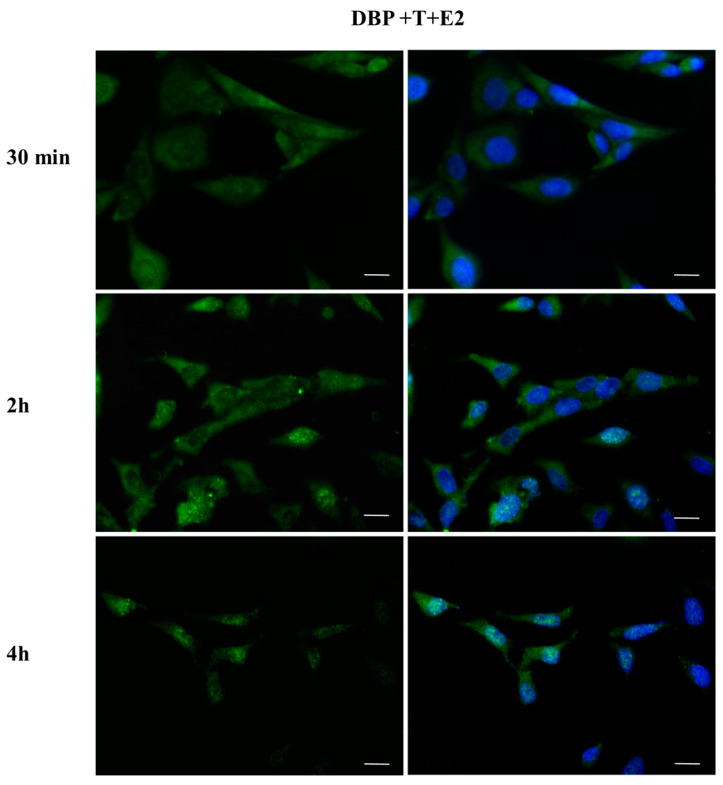
Localization of ERα after 30 min, 2 h, and 4 h of exposure to DBP + T + E2. The mixture induced cytoplasm–nuclear ERα translocation after 2 h and 4 h. Alexa Fluor 488 (green) and nuclear staining (Höechst-blue) were analyzed by immunofluorescence. Scale bar: 10 μm.

**Figure 9 ijms-24-14341-f009:**
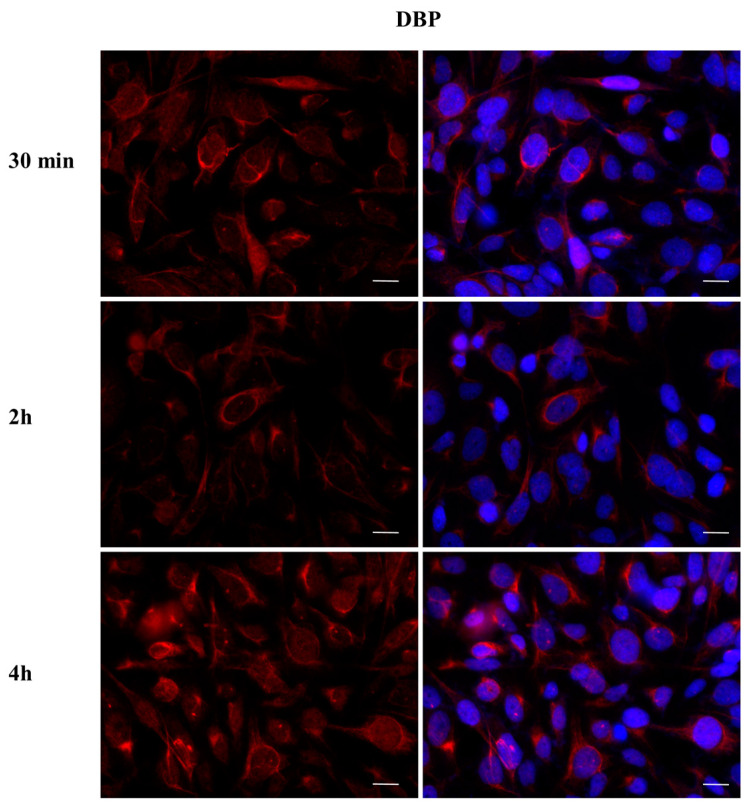
Localization of AR after 30 min, 2 h, and 4 h of exposure to DBP 10^−9^ M. DBP did not translocate AR to the nucleus in any of the treatments. Alexa Fluor 594 (red) and nuclear staining (Höechst-blue) were analyzed by immunofluorescence. Scale bar: 10 μm.

**Figure 10 ijms-24-14341-f010:**
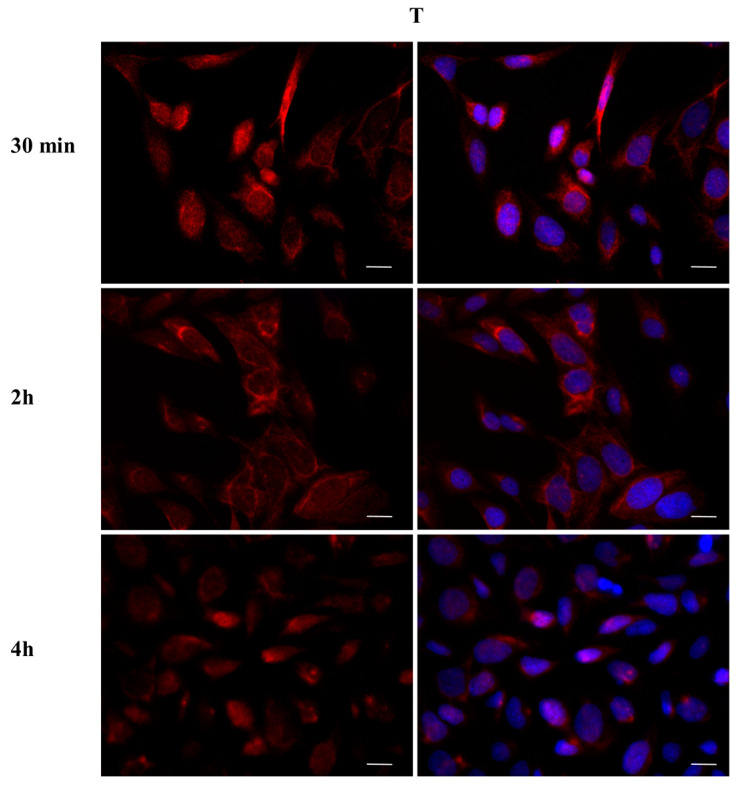
Localization of AR after 30 min, 2 h, and 4 h of exposure to T 10^−8^ M. T translocated AR to the nucleus after 30 min and 4 h of the treatments, exhibiting a cyclic behavior inside the cell. Alexa Fluor 594 (red) and nuclear staining (Höechst-blue) were analyzed by immunofluorescence. Scale bar: 10 μm.

**Figure 11 ijms-24-14341-f011:**
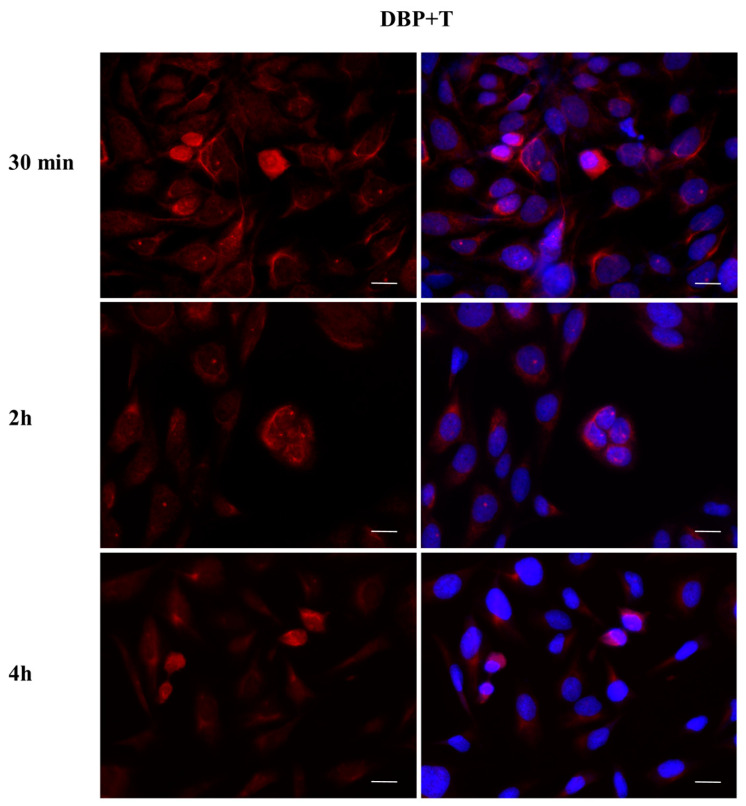
Localization of AR after 30 min, 2 h, and 4 h of exposure to DBP + T. The mixture translocated AR to the nucleus in the same way as T. Alexa Fluor 594 (red) and nuclear staining (Höechst-blue) were analyzed by immunofluorescence. Scale bar: 10 μm.

**Figure 12 ijms-24-14341-f012:**
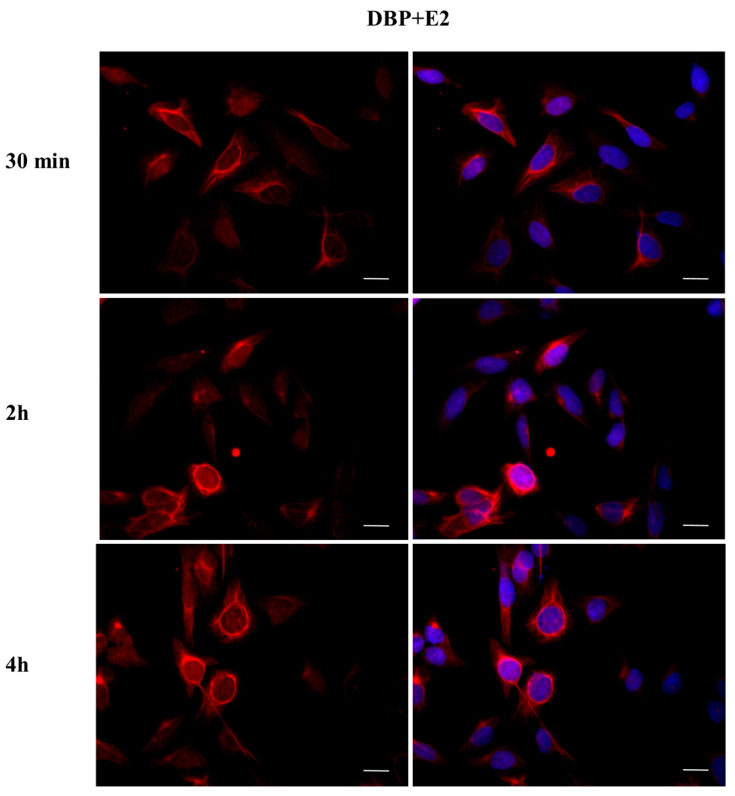
Localization of AR after 30 min, 2 h, and 4 h of exposure to DBP + E2. AR translocation did not occur, but AR is in the perinuclear area in all experimental times. Alexa Fluor 594 (red) and nuclear staining (Höechst-blue) were analyzed by immunofluorescence. Scale bar: 10 μm.

**Figure 13 ijms-24-14341-f013:**
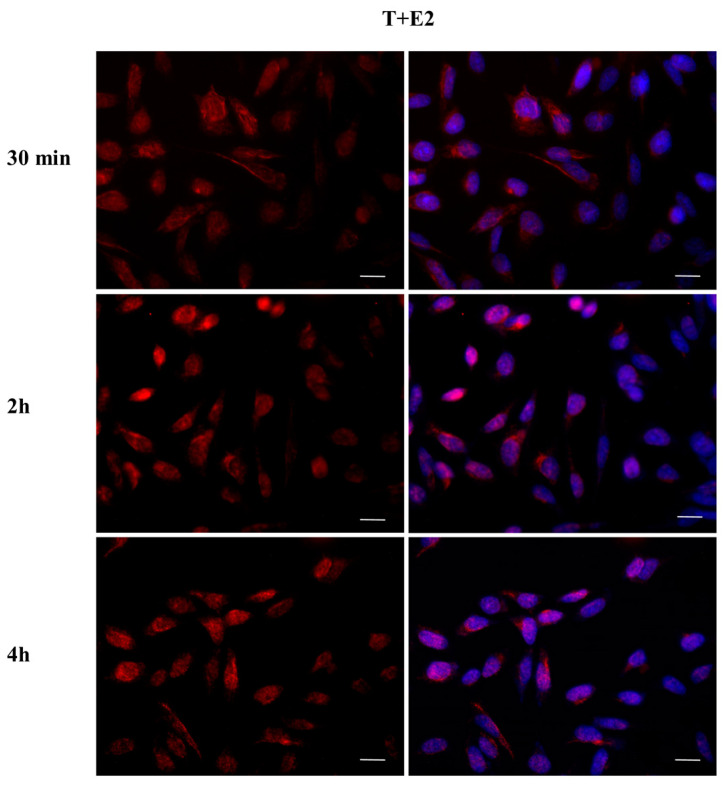
Localization of AR after 30 min, 2 h, and 4 h of exposure to T + E2. The mixture induced AR translocation in all time treatments. Alexa Fluor 594 (red) and nuclear staining (Höechst-blue) were analyzed by immunofluorescence. Scale bar: 10 μm.

**Figure 14 ijms-24-14341-f014:**
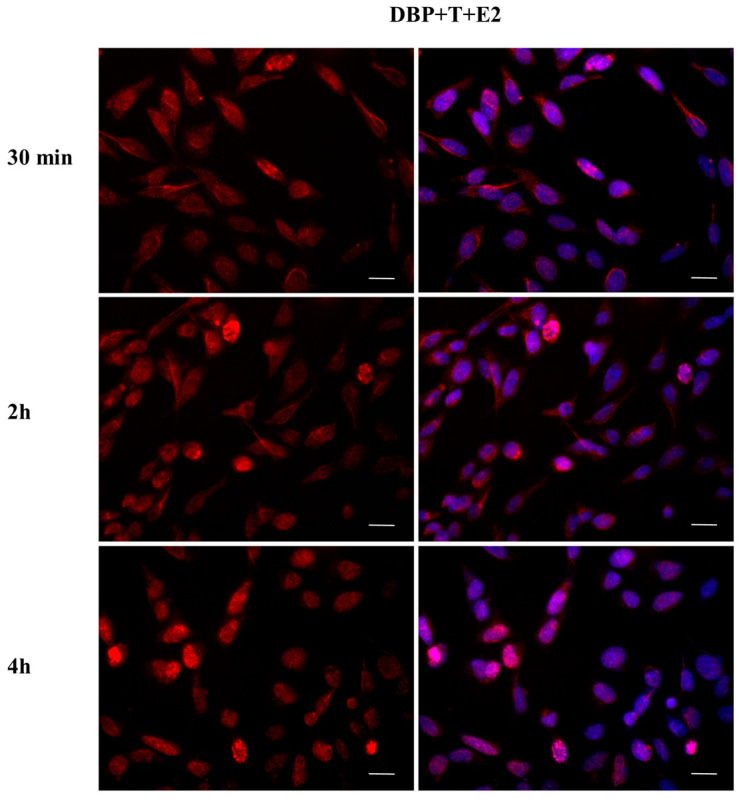
Localization of AR after 30 min, 2 h, and 4 h of exposure to DBP + T + E2. The mixture induced AR translocation in all time treatments. Alexa Fluor 594 (red) and nuclear staining (Höechst-blue) were analyzed by immunofluorescence. Scale bar: 10 μm.

**Figure 15 ijms-24-14341-f015:**
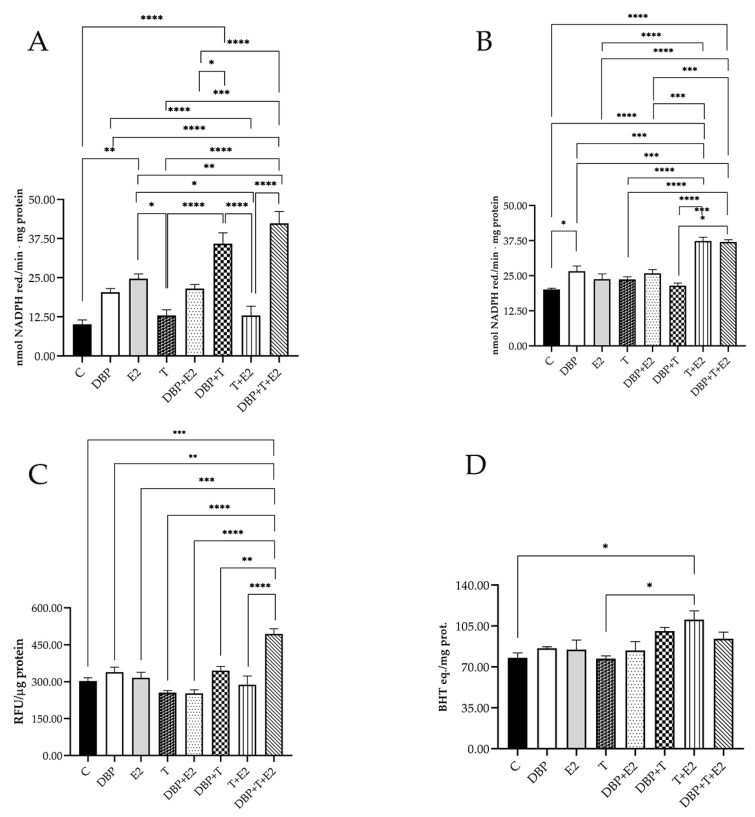
Effect of treatments on PNT1A cells redox homeostasis. Lipid hydroperoxides (**A**), susceptibility to oxidative stress (**B**), total ROS (**C**), and cell soluble antioxidant capacity (**D**) (Dunnett test: * *p* < 0.05; ** *p* < 0.01; *** *p* < 0.001; **** *p* < 0.0001).

**Figure 16 ijms-24-14341-f016:**
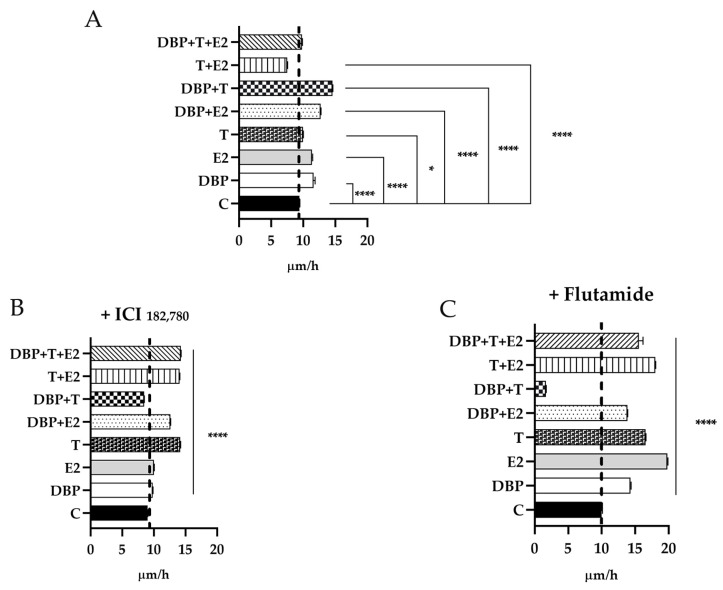
Wound healing assay performed after 24 h of exposure to T, E2, and DBP alone and in mixture. There was an increase in cell migration in all experimental classes except for the mixture T + E2 (**A**). After ICI _182,780_ treatment, DBP and E2 alone did not affect cell migration (**B**). After flutamide treatment, except for the mixture DBP + T, all the experimental classes were able to increase cell migration, with a peak after E2 and T + E2 treatments (**C**). The assay was performed using July stage, which allowed us to follow the scratch in time lapse. Average rate of healing was obtained as the ratio between space and time. Details are in the text (Dunnett test, * *p* < 0.05; **** *p* < 0.0001).

## Data Availability

Data is contained within the article.

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
