# Peer review of "Effects of Dibutylphthalate and Steroid Hormone Mixture on Human Prostate Cells"

_ijms, 2023, doi:10.3390/ijms241814341_

Round 1
Reviewer 1 Report
jms-2577423
Effects of dibutylphthalate and steroid hormone mixture on human prostate cells
Summary: The authors of this study showed that DBP has reproductive toxic effects that can cause testicular injury and reproductive system abnormalities. The effects of DBP alone and in combination with testosterone (T), estradiol (E2), and both on normal human prostate cells (PNT1A) were demonstrated by the writers in this work. Earlier studies (doi: 10.1016/j.etap.2018.08.007 and doi.org/10.1016/j.tox.2009.06.011) established the impact of DBP on the prostate. The novelty of this investigation raises questions.
Comments:
1. In Fig 1a: It is not clear why only one dose of Testosterone increases the cell viability?
2. Fig-1d: authors claim that T and DBP treatment showed greatest effects on cell viability but statistically they did not show comparison.
3. Only using one cell line is not enough to prove the hypothesis.
4. Fig-2: There is a lack of explanation why authors chose 0.5,2,4 hr time points for these experiments. How different time points effect the AR and ER expression in presence of T and DBP?
5. Is this AR and ER expression changes is limited to protein expression or DBP can effect the ER and AR mRNRA expression?
6. In IF data indicated that DBP treatment induced the nuclear localization of ERa but DBP+T treatment nuclear localization process has been inhibited. How DBP mediated the nuclear localization of ERa promotes prostate cancer cell viability? And why DBP+T treatment not prompting nuclear localization but cell viability was higher?
7. Lack of in-vivo study and mechanistic approaches to validate authors hypothesis.
Results should be more explanatory.
Reviewer 2 Report
The authors evaluate the effect of enviromental chemicals on prostate cells alone and in the presence of sex hormones. The work is of importance and demonstrates the potentially harmful effects of the Phtalates but it also adds to the fairly recently gained knowledge about estrogen receptors and their role in the pathogenesis of prostate cancer. The results support the conclusion an the methods used answers the research question adequately. Some revisions are nessecary prior to publication.
major comments:
1. Revise the outline of the manuscript: Remove content from the discussion (and add some of it to introduction). Place methods before results. Refrain from rating your results (eg "intriguingly") in the results-section but emphasize it more in the discussion section. Put your results (very briefly) in a wider context enviromental epidemiological etc.
Minor comments:
line 20: remove "that"
line 26: in a different way or differently? but not "in different"
line 52: Why does this explain the growing interest - remove or elaborate.
line 93-94: Statement lacks a reference
line 106: Reference 54 does not support your statement (about food and alcohol, on the contrary). Remove or change.
line 335: Perhaps "including" breast cancer, which is according to my knowledge also often hormone dependent.
line 346-347: Should be in the method section
line 381-413: Introduction
Generally:
Explain (to a broader group of readers the concept cell viability and why it matters.
Not a native speaker so i will refrain from commenting apart from above.
Round 2
Reviewer 1 Report
Authors addressed the all questions.
Accept in present format
